# Modeling Uplift from Observational Time-Series in Continual Scenarios

**Sanghyun Kim[1], Jungwon Choi[1], Namhee Kim[2], Jaesung Ryu[3], Juho Lee[1]**

[1]Kim Jaechul Graduate School of AI, KAIST
[2]Department of Digital Analytics, Yonsei University
[3]AFI Inc.
{nannullna, jungwon.choi, juholee}@kaist.ac.kr, cocopops@yonsei.ac.kr, kade@afidev.com

## Abstract

As the importance of causality in machine learning grows, we expect the model to learn the correct causal mechanism for robustness even under distribution shifts. Since most of the prior benchmarks focused on vision and language tasks, domain or temporal shifts in causal inference tasks have not been well explored. To this end, we introduce **Backend-TS** dataset for modeling uplift in continual learning scenarios. We build the dataset with CRUD data and propose continual learning tasks under temporal and domain shifts.

## Introduction

Uplift modeling is a particular type of predictive causal model with broad applications in marketing, personalized medicines, and politics. *Uplift* is defined as Individual Treatment Effect (ITE), but its evaluation metric differs from the other causal tasks (Radcliffe and Surry 1999). Separating causality from spurious relationships and precise estimation of treatment effects are crucial in causal tasks. However, by modeling uplift with causality, $p(y|do(t), x)$, we ultimately target a subgroup of individuals with high uplift scores, and therefore, the model's performance is measured by cumulative uplift across the population. Identifying this subgroup cannot be answered by the propensity model, $p(y|t = 1, x)$, which merely predicts one's future behavior.

In practice, the bottlenecks of causal models are data availability, scalability, and distribution shifts. In randomized controlled trials (RCTs), an individual's treatment is randomly assigned; therefore, we can *identify* Average Treatment Effect (ATE) with the difference between the treatment and control group's average outcomes (Pearl 2010). In many cases where RCTs are infeasible, however, practitioners are given observational data. No matter how many variables one has collected, unobserved confounders may still exist. Even if one can collect more covariates, the curse of dimensionality may occur. It is problematic, particularly for causal inference with high-dimensional data, as the chance of violating the positivity assumption increases (Zhao, Small, and Ertefaie 2017; D'Amour et al. 2021). Moreover, distribution may change over time and among different domains, resulting in improper validation and, eventually, the degradation of the fitted model.

To challenge the aforementioned issues with causality in high-dimensional spaces and bridge the gap between research and practice environments, we publish **Backend-TS** dataset[1], a real-world uplift dataset from mobile game users. The task is to predict uplift to push notifications by recognizing patterns from each user's CRUD[2] history. A model must learn underlying causal mechanisms and continuously adapt to distributions varying over time and to other games; otherwise, its performance will drop sharply when the distribution changes. We also argue that distribution changes can cause severe problems in causal inference since we model future customer behaviors based on their history. To the best of our knowledge, **Backend-TS** is the first uplift dataset with time-series under domain shifts.

## Background

**Causal inference and its notations.** Potential outcomes framework (Rubin 1974) defines causal effect as the difference between two potential outcomes $Y(1) - Y(0)$: when receiving treatment ($T = 1$) and under control ($T = 0$). The fundamental problem of causal inference (Holland 1986) states that either $Y_i(1)$ or $Y_i(0)$ is observable for each unit indexed by $i \in \{1, \ldots, n\}$, and the unobserved outcome is called *counterfactual*. To estimate ITE, or uplift, $u_i := Y_i(1) - Y_i(0)$, we model Conditional Average Treatment Effect (CATE) conditioned on features $\mathbf{X}$, *i.e.*, $u(\mathbf{X}) := \mathbb{E}[Y(1) - Y(0)|\mathbf{X}]$. Among the assumptions needed to identify CATE, two assumptions are crucial and often likely to be violated (Pearl 2010; Neal 2020): unconfoundedness, *i.e.*, $Y \perp\!\!\!\perp T|\mathbf{X}$, and positivity, *i.e.*, $P(T|\mathbf{X} = \mathbf{x}) > 0, \forall \mathbf{x} : P(\mathbf{X} = \mathbf{x}) > 0$.

**Uplift modeling.** Here, we introduce marketing terms following Radclifte and Simpson (2008) to illustrate the concept of uplift modeling. Individuals can be segmented into four groups along two axes: **received treatment** and **response to it**. *Sure Things* will stay (or buy a product)

---

[1]The dataset is available under CC BY-NC-SA 4.0 license at https://blog.thebackend.io/research/backend-ts, and the baseline code for models and dataloader is available at https://github.com/nannullna/ts4uplift

[2]CRUD refers to the four functions necessary for storage and server applications: create, read, update, and delete.

whether or not they receive treatment (*e.g.,* an advertisement), and *Lost Causes* will leave (or not buy the product) in either case. In short, the treatment has neither positive nor negative effects on both groups, *i.e.*, $u_i = Y_i(1) - Y_i(0) \approx 0$. On the other hand, *Persuadables* are likely to stay *only if* they receive the treatment, *i.e.*, $u_i > 0$, but *Sleeping Dogs* would be annoyed and eventually leave, *i.e.*, $u_i < 0$. Based upon this fundamental segmentation, the main goal is thus to identify as many *Persuadables* as possible while avoiding *Sleeping Dogs* for the treatment.

**Time-series modeling.** Time-series is a sequence of discrete-time data. Many previous works have dealt with regular time-series, but in this paper, we mainly focus on irregular time-series, where intervals between two consecutive data points are not the same. RNNs (Rumelhart, Hinton, and Williams 1986; Hochreiter and Schmidhuber 1997; Cho et al. 2014), TCNs (Bai, Kolter, and Koltun 2018) with dilated convolutions (Yu and Koltun 2015), and Transformers (Vaswani et al. 2017) have become popular choices for handling time-series data. However, there is no *one-size-fits-all* augmentation strategy in various types of time-series (Yue et al. 2022) except for dropout (Srivastava et al. 2014), or random masking (Devlin et al. 2018; He et al. 2022).

**Continual learning.** Continual Learning (CL) aims to effectively learn new tasks and adapt a model to distribution shifts over time while minimizing performance degradation in the learned scenarios, which is called *catastrophic forgetting* (McCloskey and Cohen 1989; Kirkpatrick et al. 2017). It is also infeasible in practice to fully retrain the model whenever new data are available due to training costs or the unavailability of previous data. Therefore, recent algorithms for CL aim to accumulate knowledge and reuse them in future scenarios without forgetting information (*e.g.*, iCaRL (Rebuffi et al. 2017), A-GEM (Chaudhry et al. 2019), EWC (Kirkpatrick et al. 2017), SI (Zenke, Poole, and Ganguli 2017)). Moreover, causal inference tasks require the model to capture the causal mechanism over distributional shifts, on which existing CL algorithms have not focused.

## Previous Benchmarks

**Benchmarks for uplift modeling.** Researchers on uplift have relied on (semi-)synthetic data for testing algorithms since underlying causal mechanisms are fully specified and counterfactuals thus exist. On the other hand, as of now, the largest observational benchmark is **Criteo dataset** (Eustache et al. 2018) with 12 static features from ~14M real-world users. Thus far, there has been little motivation to use deep learning, and therefore, related works have been restricted to smaller neural networks (# params < 1K) or other machine learning algorithms. With regard to causal inference with time-series, a subset of **MIMIC II/III** (Johnson et al. 2016) has been used for causal discovery or inference. See Moraffah et al. (2021) for a comprehensive review.

**Benchmarks for CL.** Benchmarks in various fields and tasks with CL scenarios have been introduced, *e.g.*, object recognition in robotics (Fanello et al. 2013; Lomonaco and

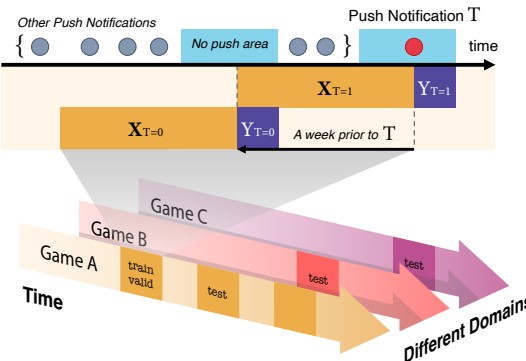

Figure 1: Illustration of **Backend-TS** dataset construction.

Maltoni 2017; She et al. 2020), classification tasks in various domains on images (Rebuffi, Bilen, and Vedaldi 2017; Lake, Salakhutdinov, and Tenenbaum 2015; He, Shen, and Cui 2021), videos (Roady et al. 2020), 3D objects (Stojanov et al. 2019), and natural language (Hussain et al. 2021; Srinivasan et al. 2022). However, domain or temporal shifts in causal inference tasks have not been well explored.

## Backend-TS Dataset

In this section, we introduce **Backend-TS** dataset. The dataset construction method and the proposed tasks are briefly illustrated in Figure 1.

**Background.** We collected data from AFI Inc., a Backend-as-a-Service (BaaS) company specializing in mobile games. The company owns backend servers and provides APIs that game developers can access to quickly release their apps without backend servers of their own. One of the features is to send **a push notification** to *all users at the same time*. We wanted to build a model that targets only a subset of users with high gains from a push message. However, CRUD log data are only available to us since the company does not collect user-specific information and has no access to each game's code or internal data.

**Construction.** The treatment is not assigned randomly in a typical observational dataset, and the true treatment assignment mechanism is unknown. In our data, however, the treatment group only exists as the push message is given to all users simultaneously. To circumvent this problem, for a train set, we sampled a *pseudo*-control group exactly one week before the push so as to eliminate the time and weekday effect. We also introduced a concept of *no push area*, an -12~+6 hour window around which no other pushes must exist to prevent interference from them. Note that some users exist in both groups, and utilizing those data points (*e.g.*, randomly choosing either one or using both) is up to modeling strategies. For a test set, we randomly split those overlapping users into either group to simulate RCTs.

**Overview.** The dataset consists of three games (A, B, and C) with a total of 16.7M lines of CRUD logs from 5,360 users. Only a handful of games met the conditions mentioned above among hundreds of games in service, most of

which either sent pushes too frequently or did not use this API at all. Each consists of a triple $(\mathbf{X}_i, T_i, Y_i)$, where $\mathbf{X}_i$ is a sequence of categorical variables along with corresponding timestamps, and $T_i, Y_i \in \{0, 1\}$ are binary indicators of the treatment and whether a gamer logged in within {three, six, twelve}[3] hours after the push message had been sent. Although the games use the same APIs provided by the company, they differ in response rates, lengths, API usage, and other factors. For example, two different games may use the same type of API calls for different purposes.

**Tasks.** We experimented on uplift modeling in the following proposed tasks:

- In-domain (ID): train with the game A (APR, MAY) and test on 20% random-split holdout set.
- Temporal shift (TS): train with the game A (APR, MAY) and test on the game A (JUN).
- Out-of-domain (OOD): train with the game A (APR, MAY) and test on the game B with fine-tuning (OOD w/) or on the game C without fine-tuning (OOD w/o).

## Experiments

| Model | Ckpt | ID | TS | OOD w/ | OOD w/o |
|---|---|---|---|---|---|
| Dragon | VAL | .091/.056 | .006/.003 | .118/.038 | .037/.023 |
| | MAX | | .112/.074 | .372/.082 | .123/.081 |
| Siamese | VAL | .145/.062 | -.036/-.011 | .154/.057 | -.057/-.030 |
| | MAX | | .249/.067 | .207/.075 | .036/.022 |
| $P(Y=1)$ | | 11.9% | 12.2% | 5.9% | 22.4% |

Table 1: Baseline results. VAL denotes the best checkpoint on the holdout set, and MAX denotes the best metric during entire training, showing the discrepancy of the performance.

**Baselines.** We used Dragonnet (Shi, Blei, and Veitch 2019) and Siamese network (Mouloud, Olivier, and Ghaith 2020) with 11 TCN blocks (receptive field of length 2,048, and each time-series was truncated accordingly.) and applied EWC for CL. Dragonnet is trained to directly predict a conditional mean $\mathbb{E}[Y|T, \mathbf{X}]$ as well as the propensity score, $e(\mathbf{X}) := P(T = 1|\mathbf{X})$, based on its sufficiency for adjustment (Rosenbaum and Rubin 1983). For Siamese network, a variable transformation method, $Z_i = \frac{T_i Y_i}{e(\mathbf{X}_i)} - \frac{(1-T_i)Y_i}{1-e(\mathbf{X}_i)}$, was used based on the fact that its conditional expectation, *i.e.*, $\mathbb{E}[Z|\mathbf{X}]$, is equal to the true uplift $u(\mathbf{X})$ (Athey and Imbens 2015). We attached an embedding layer with Layer-Norm (Ba, Kiros, and Hinton 2016) which is similar to language models like BERT (Devlin et al. 2018) for categorical variables and used sinusoidal functions to encode second, hour and weekday information as follows:

$$f(t) = \left[ \sin\left( \frac{2\pi t}{\max_t} \right), \cos\left( \frac{2\pi t}{\max_t} \right) \right],$$

where $\max_t$ is the maximum possible value of $t$, *i.e.*, 3600 seconds in an hour, 24 hours in a day, and 7 for weekday.

---

[3]The shorter the time interval, the greater the influence of the push, but the smaller the number of people responding. In our experiment, "three hours" was used as a target.

**Evaluation.** The performance of an uplift model can be evaluated by qini coefficients (QINI) (Radcliffe 2007) and area under uplift curve (AUUC) (Devriendt et al. 2020). The two metrics are basically similar, measuring cumulative incremental gains when the treatment is given only to the top individuals sorted by uplift scores predicted by the model.

**Results.** Table 1 shows QINIs (left) and AUUCs (right) of the best checkpoint on the holdout set (VAL) and among the entire training checkpoints (MAX) for each task. The difference between VAL and MAX can be attributed to the model capturing spurious correlations rather than the true mechanisms and the wrong validation due to distributional shifts.

- TS: The performance gap between VAL and MAX was significant, and VAL actually performed worse than random targeting (QINI & AUUC below zero). This empirically shows the existence of the temporal distribution changes.
- OOD W/: Fine-tuning with the additional data using the CL algorithm has somewhat reduced the performance gap. We conjecture that the model became more robust since it further learns common mechanisms and forgets relationships irrelevant to the true effect.
- OOD W/O: The performance dropped sharply without fine-tuning. We emphasize that the true causal model should perform equally well both in ID and TS and generalize to different games even without training, although they may potentially have a very different user base.

## Conclusion and Future Work

In this paper, we introduce **Backend-TS** dataset and propose uplift tasks accordingly, combining causal inference with CL scenarios. We demonstrate that naïvely applying existing methods may fail as uplift modeling tries to predict future behaviors based on historical data. All observational datasets have inherent biases; identifying causal relationships and eliminating undesirable effects would be one of the most important follow-up research topics. We believe that learning causal mechanisms invariant over time is crucial for the way toward general-level AI and that the dataset will contribute to developing such algorithms.

## Ethical Statement and Societal Impact

We did not collect any sensitive information, and all data have been fully anonymized. Do not attempt to misuse it for purposes other than research, including but not limited to, identifying individuals or games, hacking, and cracking the system. **Backend-TS** will contribute to developing robust models and algorithms that can infer correct causal mechanisms in high-dimensional spaces.

## Acknowledgement

We thank AFI Inc. and anonymous game companies for allowing data to be published for research purpose. This work was supported by Institute of Information & communications Technology Planning & Evaluation (IITP) grant funded by the Korea government(MSIT) (No.2019-0-00075, Artificial Intelligence Graduate School Program(KAIST)).

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
