# OpenReview forum: "Modeling Uplift from Observational Time-Series in Continual Scenarios"
_AAAI.org/2023/Bridge/CCBridge — AAAI23 Bridge Continual Causality_

### Official Review · Reviewer_1D6V · 2022-11-28
**a starting point towards common causal-continual benchmarks**

**Rating:** 7
**Confidence:** 3

**Review:**

The paper proposes a novel dataset that contains casual relationship over time and over multiple domains. The data is about push notifications of three different games.

From a continual learning perspective, I find the dataset interesting and well made, although there are not enough details to give a full evaluation. I also think that a dataset is a good contribution for the bridge since having common benchmarks is a good way to start exploring the two fields together.

The paper claims that:
> We conjecture that it is due to the model being overfitted to the train set without learning causal mechanisms, and a model should perform equally well both in ID and OOD.

Which I find a bit surprising. Personally, I would not expect the performance of the model to generalize to different apps without training since they may potentially have a very different user base.
However, I tend to agree with the general point that causal learning may help to generalize OOD.

---

### Official Review · Reviewer_JJrw · 2022-12-01
**A new dataset with benchmark performance**

**Rating:** 8
**Confidence:** 3

**Review:**

One problem of bridging two fairly separate fields is the standardization of benchmarks. For that we need datasets with annotations that are understandable for both fields. UpliftCRUD is trying to provide that. So, thematically speaking, it's a great match.
There are minor typos that need to be fixed. For example, "Sure Things will stay whether or not they receive treatment ...". I guess you meant "say".

---

### Official Review · Reviewer_LZSP · 2022-12-05
**Interesting Dataset for Causal Inference in Non-Stationary Environments**

**Rating:** 7
**Confidence:** 3

**Review:**

The paper introduces a new interesting dataset called "UpliftCRUD" for addressing causal inference within CL scenarios.

Pro:
* The paper is well focused and organized, within the scope of the workshop.
* The paper proposes an interesting new dataset composed of 16.7M lines of CRUD logs from 5,360 users, which seems very significant.
* The paper demonstrates that naıvely applying existing methods may fail as uplift modeling tries to predict future behaviors based on
historical data.

Cons:
* I would have appreciated a better introduction to "Uplift" a "CRUD". Current "Introduction" should be significantly improved.

Overall, I believe this paper to be a good addition to the workshop.

---

### Decision · Program_Chairs · 2022-12-05

**Decision:**

Accept

**Comment:**

Accept - Oral

This paper proposes a novel benchmark, “UpliftCRUD” poised to bridge both of the fields of continual learning and causal inference. Novel benchmarks are one of the key focus areas of the bridge, and thus it fits the theme well and addresses a critical need. We suggest that the authors integrate the reviewers' comments for the camera-ready version, including improving the introduction substantially to explain the background concepts, as well as substantiate the paper’s claims of model generalization to different apps without training. Minor typos should also be corrected.